# Evaluation Pitfalls in Data Augmentation for Adversarial Robustness

## Abstract

Recent work has proposed novel data augmentation methods to improve adversarial robustness of deep neural networks. In this paper, we re-evaluate such methods under a common framework and through the lens of different metrics that characterize the augmented manifold, finding contradictory evidence. In particular, our extensive empirical analysis involving 5 data augmentation methods, tested with 10 augmentation probabilities, shows that: (i) novel data augmentation methods proposed to improve adversarial robustness only improve it when combined with classical augmentations, like image flipping and rotation; (ii) novel data augmentation methods even worsen adversarial robustness if not combined with classical augmentations; and (iii) adversarial robustness is significantly affected by augmentation probability, conversely to what claimed in recent work. We conclude by discussing how to rethink the development and the evaluation of novel data augmentation methods for adversarial robustness.

## 1 Introduction

Data augmentation (DA) is a widely-used technique that applies randomly constructed transformations on the input data to increase the diversity and size of the training set. The underlying rationale is that for better generalization and performance of machine learning (ML) algorithms, more data is needed (Schmidt et al., 2018; Nakkiran et al., 2020; Rebuffi et al., 2021). More specifically, it has been shown that DA can have a regularization effect for some combinations of augmentation and ML methods, including regression (Bishop, 2007), kernel methods (Dao et al., 2019), and deep learning (Chen et al., 2020). Additional potential positive effects of DA consist of reducing dataset bias (McLaughlin et al., 2015), improving accuracy (Ciregan et al., 2012; Krizhevsky et al., 2012), and enhancing algorithmic fairness (Sharma et al., 2020).

Furthermore, Rebuffi et al. (2021) have shown that newly-proposed *heuristic* DA methods like MixUp (Zhang et al., 2017), CutMix (Yun et al., 2019), ManifoldMixUp (Verma et al., 2019), and CutOut (DeVries & Taylor, 2017), as well as *generative* DA methods like Diffusion Models (Ho et al., 2020) are able to improve *adversarial robustness*, namely, the ability of the model to withstand *adversarial examples* (i.e., maliciously-perturbed inputs aimed to mislead classification) (Dalvi et al., 2004; Szegedy et al., 2014). However, all these approaches have been tested in combination with classical augmentations (e.g., rotation, flipping, color-jittering), and using a fixed fraction of augmented samples (i.e., a fixed *augmentation probability* choice).

It thus remains an open question to understand whether the claims made in previous work hold. For this reason, we first formulate the following two working hypotheses, based on empirical evidence from prior work.

---

**Hypothesis 1.** *Newly-proposed heuristic and generative DAs increase adversarial robustness besides classical augmentations (DeVries & Taylor, 2017; Guo et al., 2019; Verma et al., 2019; Yun et al., 2019; Nakkiran et al., 2020).*

---

> **Hypothesis 2.** *The percentage of augmented samples does not significantly influence generalization and adversarial robustness (Krizhevsky et al., 2009; DeVries & Taylor, 2017; Guo et al., 2019; Verma et al., 2019; Yun et al., 2019; Nakkiran et al., 2020).*

In this work, we shed light on these questions by first reviewing and categorizing DAs, as well as the conditions under which these DAs have been tested (Section 2). We then propose a unifying framework to re-evaluate such augmentation techniques through the lens of different metrics that characterize the augmented manifold, aiming to verify the aforementioned claims (Section 3). Our evaluation framework consists of three main components: (i) a performance-vs-robustness analysis of DAs, which allows us to decouple the impact of heuristic, data-driven, and classical augmentations on adversarial robustness, as well as to understand how different augmentation probabilities affect it; and two additional metrics named (ii) decision-function roughness, and (iii) data-augmentation spuriousness, which provide additional insights on how DAs impact adversarial robustness.

Our extensive empirical analysis, involving 5 DA methods tested with 10 different augmentation probabilities, shows contradictory evidence with respect to previous studies (Section 4). In particular, we find that:

1. both novel heuristic and generative DA methods claimed to improve adversarial robustness only improve to a significant level, when combined with classical augmentations;

2. novel DA methods even worsen adversarial robustness if not combined with classical augmentations;

3. and adversarial robustness is significantly affected by augmentation probability, conversely to what claimed in recent work.

We conclude the paper by discussing why we believe that our findings are relevant towards designing novel DA methods for adversarial robustness (Section 5). The main reason is that the reported contradictory evidences from prior work demand for the adoption of a proper evaluation framework and a common benchmark for DA methods, especially when it comes to evaluating their adversarial robustness, and we firmly believe that our work provides an important first contribution in this direction.

## 2 Data Augmentation Methods

In this section, we present background in DA and review related works. We first discuss different DA techniques, starting from the classical approaches, like rotation or cropping, in vision. Afterwards, we describe heuristic augmentations, that are domain independent, like mixing several samples from a dataset. Next, we discuss data-driven augmentations which are based on learning strategies such as generative models. To conclude the section, we review existing works that connect DA with adversarial robustness.

**Classical Augmentations**: Incorporating domain knowledge of experts in models by using DA has been one of the main approaches to improve performance and non-adversarial robustness. From the early days of deep learning, simple geometrical transformations have been utilised as data augmentation with great success Krizhevsky et al. (2012). For example, in the vision domain, horizontal flipping, random rotations, as well as slight change in brightness, contrast and saturation of natural images were used. In particular the latter simulate different camera angles and lighting conditions which are known to preserve the main characteristics of the data w.r.t. the task at hand (Krizhevsky et al., 2009; He et al., 2016).

**Heuristic Augmentations**: The design of classical DAs requires domain knowledge and deep understanding of the task and data at hand. Several efforts have been made in order to introduce DAs based on more general heuristics that are domain or task independent, as opposed to classical augmentations. For example MixUp (Zhang et al., 2017) is a heuristic-based augmentation that creates new samples by linearly combining existing data and their labels, resulting in better generalisation and improved adversarial and non-adversarial robustness. CutOut (DeVries & Taylor, 2017) instead removes certain areas of the input in order to create new samples, and thus increases classification performance and resilience against missing data. CutMix (Yun et al., 2019) combines the two previous heuristics, and generates new data by mixing cut-out regions into

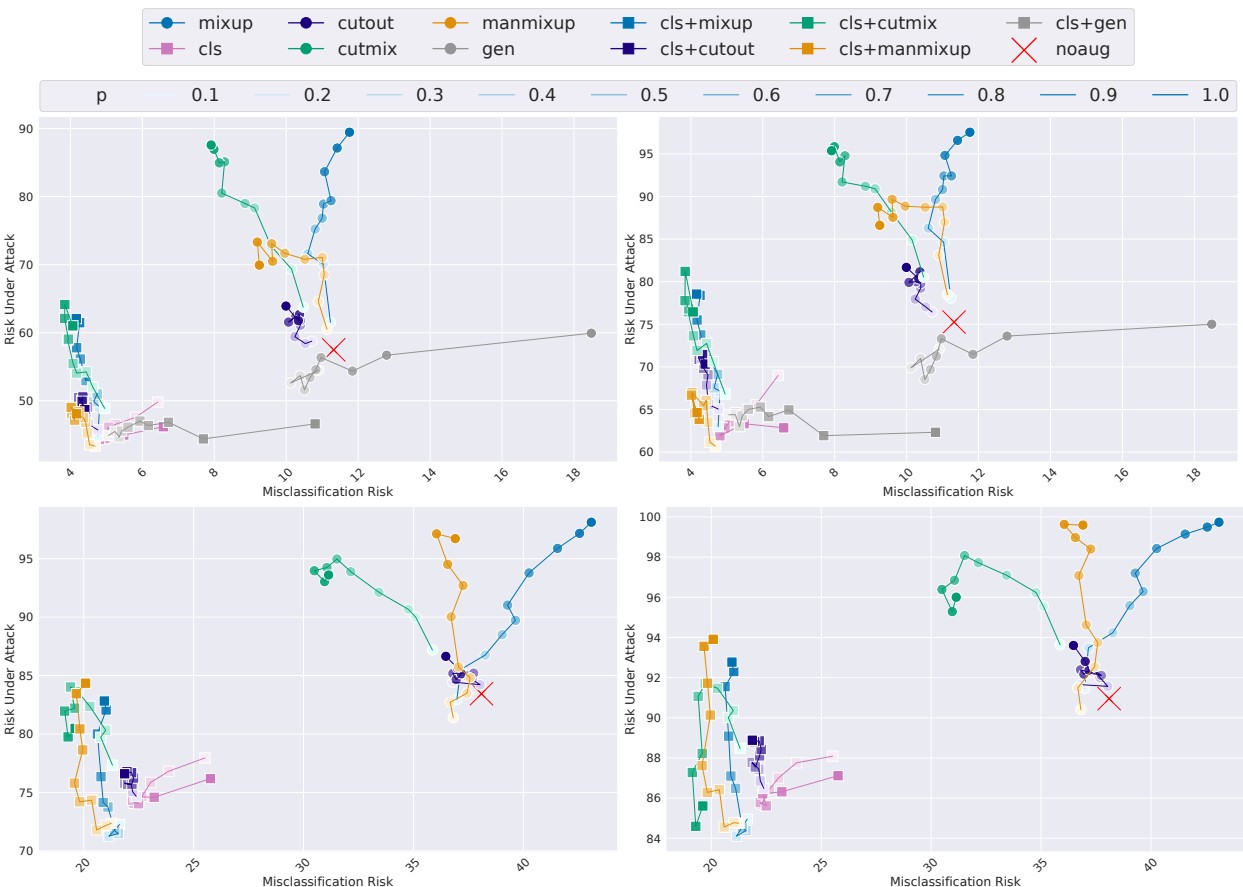

Figure 1: Robustness vs. Performance. First column: risk under attack ($L_2$,$\mathrm{PGD}_\epsilon = 0.1$) VS. misclassification risk. Second column: risk under attack ($L_\infty$,$\mathrm{PGD}_\epsilon = \frac{1}{255}$) vs. misclassification risk. First row: CIFAR10. Second row: CIFAR100.

existing samples, which improves out-of-distribution generalisation. Finally, Manifold-MixUp (Verma et al., 2019) not only mixes data and labels, but does so on the intermediate representations of the neural network. Manifold-MixUp leads to increased adversarial and non-adversarial robustness and generalisation.

**Data-Driven Augmentations**: Another perspective on DA is to use existing data to *learn* suitable transformation strategies. Generative models such as GANs (Goodfellow et al., 2014), VAEs (Kingma & Welling, 2013), and Denoising Diffusion models (Ho et al., 2020) have recently been popular models for DA (Antoniou et al., 2017; Bowles et al., 2018; Child, 2020; Nakkiran et al., 2020). However, Rebuffi et al. (2021) have shown that the latter diffusion models are more successful in terms of generalisation and adversarial robustness. In addition to generative models, other data-driven approaches exist. For example Augerino (Benton et al., 2020) learns affine transformations from data within the borders of robust augmentations. Another example is AutoAugment (Cubuk et al., 2018), where reinforcement learning is used to fine-tune augmentation hyperparameters.

**DA and Robustness**: The relation between DA and adversarial robustness is subject of ongoing research. We attempt to provide an overview in Table 1. Many recently-proposed heuristic and data-driven DAs were shown to increase adversarial robustness (Zhang et al., 2017; Verma et al., 2019; Yun et al., 2019; DeVries & Taylor, 2017). However, at the same time, there are inductive biases in these studies. In all cases, the studied DA

1. is combined with classical augmentations,

Table 1: DA approaches and their effect on robustness and accuracy. For each DA, we report whether it is shown to improve ($\uparrow$), worsen ($\downarrow$), or not affect ($-$) clean accuracy (Acc.), and robustness (Rob.) to attacks (ADV, either FGSM or PGD), corruptions (COR) or deformations (DEF). We also report if the proposed DA is combined with classic augmentation (cls), adversarial training (AT), and the augmentation probability (P) used, where 'lin' denotes the linear strategy used by CutMix to increase the augmentation probability throughout training.

| Reference | DA | Rob. | Acc. | cls | P | AT |
|---|---|---|---|---|---|---|
| Zhang17 | MixUp (heuristic) | $\uparrow$ ADV | $\uparrow$ | ✓ | 1. | |
| Verma19 | | $\uparrow$ ADV | $\uparrow$ | ✓ | 1. | |
| Verma19 | | $\uparrow$ ADV | $\uparrow$ | ✓ | 1. | |
| Yun19 | | $\uparrow$ ADV | $-$ | ✓ | 1. | |
| Guo19 | | $-$ | $\downarrow$ | ✓ | 1. | |
| Rebuffi21 | | $\downarrow$ ADV | $-$ | ✓ | 1. | ✓ |
| Yun19 | CutMix (heuristic) | $\uparrow$ ADV | $\uparrow$ | ✓ | lin | |
| Rebuffi21 | | $-$ ADV | $-$ | ✓ | lin | ✓ |
| Hendrycks19 | | $\downarrow$ COR | $-$ | ✓ | lin | |
| Devries17 | CutOut (heuristic) | $-$ | $\uparrow$ | ✓ | 0.5 | |
| Hendrycks19 | | $\downarrow$ COR | $-$ | ✓ | 0.5 | |
| Rebuffi21 | | $\uparrow$ ADV | $-$ | ✓ | 0.5 | ✓ |
| Rebuffi21 | Gen (data.) | $\uparrow$ ADV | $\uparrow$ | ✓ | 0.9 | ✓ |
| Nakkiran20 | | $-$ | $\uparrow$ | ✓ | 0.9 | |
| Verma19 | Man (heu.) | $\uparrow$ ADV | $\uparrow$ | ✓ | 1. | |
| Verma19 | | $\uparrow$ DEF | $\uparrow$ | ✓ | 1. | |

2. and is tested on only one augmentation probability choice (mostly 0.5, sometimes 1.0, and occasionally a fixed linearly increasing regime) is used.

In some works, DA is even further combined with *adversarial training*, i.e., including adversarial examples into the training set. Consequently, it is not clear which factor is really contributing to improve or degrade adversarial robustness, and to what extent. In this work, we overcome this limitation by proposing a comprehensive framework that properly assesses robustness of DAs. Our framework decouples the effect of each of the aforementioned factors, and highlights the real impact of newly-proposed heuristic and data-driven DAs on adversarial robustness.

# 3 Evaluating Data Augmentation for Adversarial Robustness

In this section, we first introduce the learning setup and notation (Section 3.1), and then present our evaluation framework consisting of three main components: (i) performance-vs-robustness analysis of DAs, which aims to decouple the effect of heuristic, data-driven, and classical augmentations, along with different augmentation probabilities, on robustness and performance (Section 3.2); (ii) decision-function roughness (Section 3.3), and (iii) data-augmentation spuriousness (Section 3.4), which aim to provide additional insights on how DAs impact adversarial robustness.

## 3.1 Learning Setup and Notation

Throughout this work, let $X$ be a random variable on a probability space $(\mathcal{X}, \mathcal{A}, P)$ with sigma algebra $\mathcal{A}$ and input space $\mathcal{X} \subset \mathbb{R}^d$, e.g. images, and denote by $P$ the probability measure of $X$. Further, let $l : \mathcal{X} \to \mathcal{Y}$ be a labeling function to a finite set $\mathcal{Y} \subset \mathbb{N}$ of labels, e.g. $\{1, \ldots, c\}$. Given a class $\mathcal{F}$ of functions $f : \mathcal{X} \to \mathcal{Y}$ and a sample $S = ((\mathbf{x}_1, l(\mathbf{x}_1)), \ldots, (\mathbf{x}_s, l(\mathbf{x}_s))) \in (\mathcal{X} \times \mathcal{Y})^s$ with $\mathbf{x}_1, \ldots, \mathbf{x}_s$ independently drawn from $P_X$, the problem of *risk minimization* is to find a function $f \in \mathcal{F}$ with low *misclassification risk* (Cucker & Smale,

2002), written formally as

$$R(f, l) := P(f(X) \neq l(X)). \tag{1}$$

One successful method for solving problems of risk minimization is to perform Stochastic Gradient Descent (SGD) based on some parametric function class $\mathcal{F}$ of neural networks (LeCun et al., 2015).

In many practical tasks, risk minimization can be improved by applying so-called *data augmentation* techniques. In this work, we call a random function $A : (\mathcal{X} \times \mathcal{Y})^s \to \left\{ X \times Y : \mathcal{X} \times \mathcal{Y} \to \mathbb{R}^d \right\}^r$ a data augmentation, if it maps the sample $S$ to some vector $A(S) = (X_1 \times Y_1, \ldots, X_r \times Y_r)$ of independent random variables $X_1 \times Y_1, \ldots, X_r \times Y_r$ with measure $P_{X_1 \times Y_1}$ on $\mathcal{X} \times \mathcal{Y}$ such that the marginal measure $P_{X_1}$ dominates $P_X$, i.e. such that the sample $S$ is included in the augmented sample $\tilde{S}$ observed from the random variable $A(S)$.

As discussed in Section 2, prior works reported that training on augmented samples leads to models with lower *adversarial risk*, when compared to models obtained without data augmentation. One classical measure for *adversarial risk* is the *risk under corrupted inputs* (Mansour & Schain, 2014; Attias et al., 2018):

$$R_{\mathrm{cor}}(f, l, \epsilon) := P(\exists \mathbf{x} \in B_\epsilon(X) : f(\mathbf{x}) \neq l(X)), \tag{2}$$

with $B_\epsilon(\mathbf{x}) := \{\mathbf{x}' \in \mathbb{R}^d \mid \|\mathbf{x}' - \mathbf{x}\| \leq \epsilon\}$. One common approach to approximate equation 2, which we follow in Section 3.2, is to apply adversarial attacks on test samples to compute the empirical expected risk. Another measure for adversarial risk is the *prediction-change risk* (Szegedy et al., 2014), i.e., the probability that a sample is classified differently within the given $\epsilon$-ball:

$$R_{\mathrm{pc}}(f, \epsilon) := P(\exists \mathbf{x} \in B_\epsilon(X) : f(\mathbf{x}) \neq f(X)). \tag{3}$$

We will propose a new approximation for equation 3 in Section 3.3.

## 3.2 Performance-vs-Robustness Analysis

We introduce here our performance-vs-robustness analysis. The underlying idea is to separately evaluate the impact on performance and robustness of newly-proposed heuristic and data-driven augmentations against classical augmentations. Furthermore, we incorporate the analysis of a range of augmentation probabilities. The goal is to understand whether such newly-proposed heuristic and data-driven augmentations are really responsible for improving adversarial robustness and performance, or to which extent, instead, classical augmentations contribute to that. Moreover, we also aim to evaluate how using different augmentation probabilities affect performance and robustness.

To this end, we propose to look at two axes at once: 1) the classification error, and 2) adversarial vulnerability. We first evaluate each newly-proposed heuristic and data-driven DA *without* using any classical augmentation, and then in combination with classical augmentations, using different augmentation probabilities. A DA technique is retained useful if it pushes the corresponding point towards the origin of this plot (i.e., towards reducing both classification error and adversarial vulnerability). As we will see in the experimental section, some of the newly-proposed heuristic and data-driven DAs, which were originally meant to improve robustness and performance, do worsen them instead.

We measure the performance of models by estimating the misclassification risk in equation 1 of a model $f$ for a set of test samples $\tilde{S} := ((\mathbf{x}'_1, l(\mathbf{x}'_1)), \ldots, (\mathbf{x}'_s, l(\mathbf{x}'_s)))$ by

$$\widehat{R}(f, \tilde{S}) := \frac{1}{s} \sum_{i=1}^s \mathbf{1}[f(\mathbf{x}_i) \neq l(\mathbf{x}_i)], \tag{4}$$

where $\mathbf{1}[a \neq b] = 1$, e.g. if the label disagrees, and $\mathbf{1}[a \neq b] = 0$ otherwise or if the label agrees. We measure the adversarial vulnerability of models by estimating the risk under corrupted inputs in equation 2. For estimating equation 2 we rely on an adversarial attack. More concretely, we compute an adversarial example for an input $\mathbf{x}^0 \in \mathcal{X}$ using Projected Gradient Descent (PGD) (Madry et al., 2017) as follows:

$$\mathbf{x}^t = \Pi_\epsilon \left( \mathbf{x}^{t-1} + \alpha \operatorname{sgn}(\nabla_{\mathbf{x}} L(f(\mathbf{x}), l(\mathbf{x}))) \right), \tag{5}$$

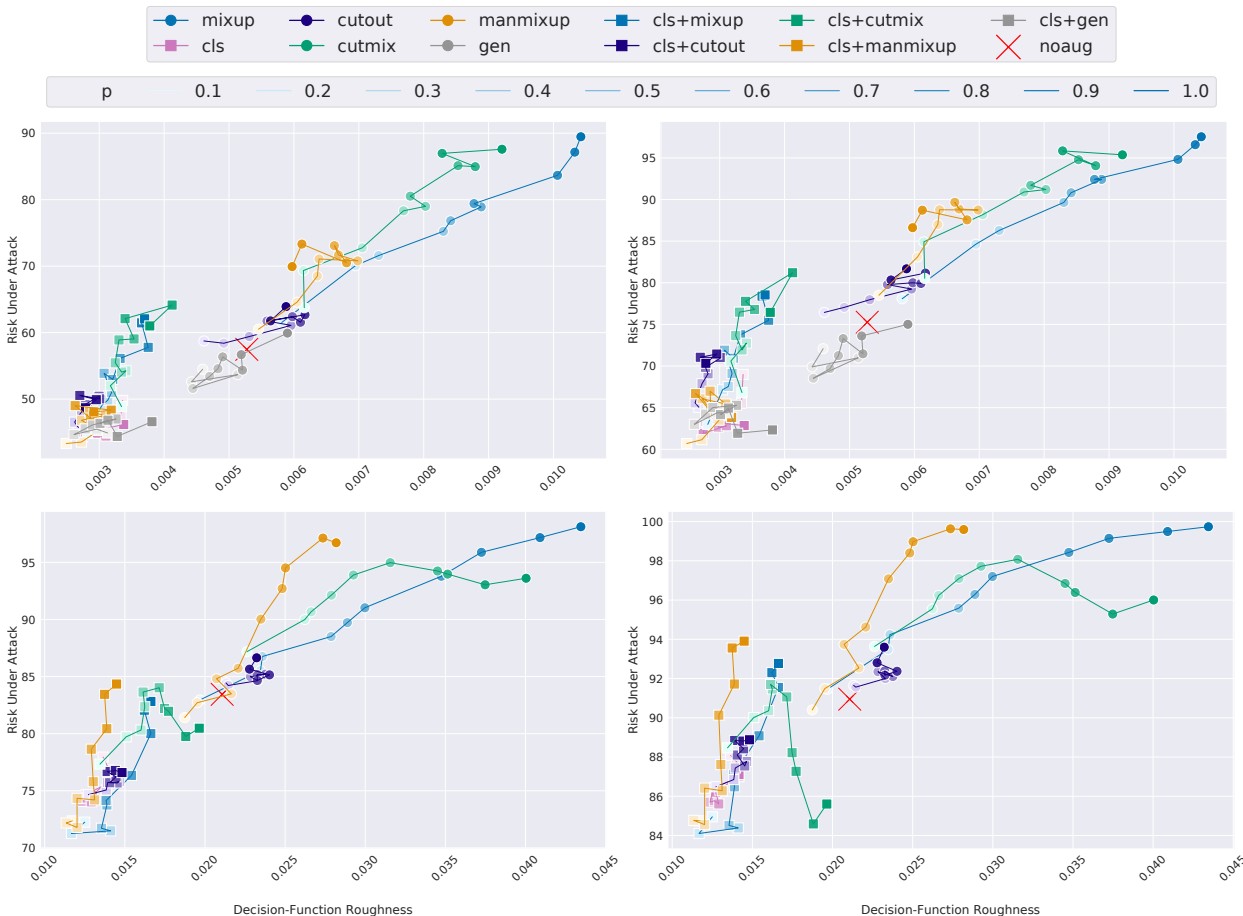

Figure 2: Decision-function roughness vs. Robustness. First column: decision-function roughness vs. risk under attack ($L_2$, $\mathrm{PGD}_\epsilon = 0.1$). Second column: decision-function roughness vs. risk under attack ($L_\infty$, $\mathrm{PGD}_\epsilon = \frac{1}{255}$). First row: CIFAR10. Second row: CIFAR100.

where $L : \mathcal{Y} \times \mathcal{Y} \to \mathbb{R}$ is a loss, $\alpha > 0$ is the step-size and $\Pi_\epsilon$ is a projection from the inputs $\mathcal{X}$ into the ball $B_\epsilon(\mathbf{x})$, and $\epsilon > 0$. [1] We then approximate equation 2 by the misclassification risk $\widehat{R}(f, \tilde{S})$ on the sample $\tilde{S} := ((\mathbf{x}_1^t, l(\mathbf{x}_1)), \ldots, (\mathbf{x}_s^t, l(\mathbf{x}_s)))$ where $\mathbf{x}_1^t, \ldots, \mathbf{x}_s^t$ are the adversarial examples computed by applying equation 5 $t$ times. We call $\widehat{R}(f, \tilde{S})$ the *risk under attack (RUA)*.

### 3.3 Decision-Function Roughness

We introduce here a measure for decision-function roughness that connects adversarial vulnerability with the shape of the decision surface learned by the model. In particular, models showing a rougher decision surface are expected to be more vulnerable to adversarial examples. We aim to evaluate adversarial robustness beyond classical estimates of the risk under attack (equation 2). We propose a new estimate for prediction-change risk equation 3, the *decision-function roughness*. On a high level, decision-function roughness is related to the shape of the decision surface of a classifier which is known to influence generalization (Arpit et al., 2017; Jin et al., 2020) and evasion robustness (Lyu et al., 2015; Cisse et al., 2017; Sokolic et al., 2017; Cohen et al., 2019).

Existing, similar measures use dimensionality reduction to apply noise (Shu & Zhu, 2019), and are based on Gaussian noise (Forouzesh et al., 2021), or rely on Gaussian noise to estimate the Jacobian of the classifier (Novak et al., 2018). Our estimate of equation 3 is instead based on uniform noise. Due to the

---

[1]We refer to the $\epsilon$ and $\alpha$ used in PGD as $\mathrm{PGD}_\epsilon$ and $\mathrm{PGD}_\alpha$

phenomenon of concentration of measure (Donoho, 2000), for high dimension $d$ the Lebesgue measure $\lambda$ in a ball $B_r^{(d)}$ of radius $r$ is concentrated at its surface. That is, for any $\delta > 0$, we have $\lim_{d \to \infty} (\lambda(B_r^{(d)}) - \lambda(B_{r-\delta}^{(d)})) / \lambda(B_r^{(d)}) = 1$ given any $L_p$-norm ($p \in [1, \infty]$) we take. As the dimension $d$ in most applications is high, the concentration of measure effect motivates to approximate equation 3 by sampling only from the shell $\partial B_\epsilon$ (points at distance $\epsilon$ from the center of the ball) instead of sampling from the full ball $B_\epsilon$. By considering only the points on the boundary $\partial B_\epsilon$ of the ball $B_\epsilon$, we introduce the *measure of decision-function roughness* on a sample $\tilde{S} := ((\mathbf{x}_1', l(\mathbf{x}_1')), \ldots, (\mathbf{x}_s', l(\mathbf{x}_s')))$ by

$$\hat{r}(f, \tilde{S}, \epsilon) := \frac{1}{s} \sum_{i=1}^{s} \frac{1}{n} \sum_{j=1}^{n} \mathbf{1}[f(\mathbf{x}_i') \neq f(\mathbf{y}_{ij})], \tag{6}$$

where $\mathbf{y}_{i1}, \ldots, \mathbf{y}_{in}$ are uniformly drawn from $\partial B_\epsilon(\mathbf{x}_i')$. Note that $\hat{r}(f, \tilde{S}, \epsilon) \approx R_{\mathrm{pc}}(f, \epsilon)$ for sufficiently high dimension $d$ and sample size $s$. In a nutshell, the rougher the decision function, the less robust is the model at point $\mathbf{x}_0$.

### 3.4 Data-Augmentation Spuriousness

To gain a better understanding on how DA affects robustness, we measure *spuriousness* as the fraction of the augmented data having their closest neighbor in a spurious set. We use the set of *non-robust features* described by Ilyas et al. (2019), which are spurious features generated using activations of a non-robust model. These are features that are highly predictive, yet meaningless for humans. These features, however, are exploited in adversarial example crafting. To put the spuriousness in perspective, as an additional dimension we also use *robust* features, which represent robust characteristics in the data space, and provide distance of augmented data to the robust features.

In order to find the distance of augmented data from the robust and non-robust manifolds, we prepare 3 sets: 1) the augmented set, 2) the set of non-robust features, and 3) the set of robust features. We then find the closest neighbor for each augmented sample among all these sets, and then calculate the percentage of augmented samples in robust and non-robust sets as a measure of an approximated distance to their manifolds. The closeness of augmented samples to such *non-robust features* is thus an indicator for the existence of spurious features in augmented data, which could be a cause for increased adversarial risk in models.

## 4 Experiments

In this section, we put our framework into practice to study DA and adversarial robustness. To this end, we ran experiments on two datasets (CIFAR10 and CIFAR100) that are ubiquitous and more crucially consistently used in the previous studies of DA, and evaluated them using our proposed framework. We assess performance-vs-robustness to address Hypotheses 1 and 2 in Section 4.2.1, decision-function roughness (Section 4.2.2), and data-augmentation spuriousness (Section 4.2.3).

### 4.1 Setup

We first describe the setup for the performance-vs-robustness study, continue with the decision-function roughness setup and conclude with the setup of the data-augmentation spuriousness experiments. We publish the code of all our experiments.[2]

**Robustness Setup:** We test robustness against PGD with $L_2$ and $L_\infty$ norms. More specifically, in Figure 1, we chose a perturbation size that is only large enough to demonstrate differences in adversarial risk of trained models.[3] Results for other configurations are in Appendix B. We further evaluate the following DA methods: MixUp, Manifold-MixUp (Man), CutMix, CutOut, and a Denoising Diffusion Probabilistic generative model (Gen). For the latter, we rely on data from Nakkiran et al. (2020) which is only provided for CIFAR10, we

---

[2]URL blinded for anonymous submission.
[3]$\mathrm{PGD}_\epsilon = 0.1$ for $L_2$ norm, and $\mathrm{PGD}_\epsilon = \frac{1}{255}$ for $L_\infty$ norm

thus are not able to include results on CIFAR100. Finally, we also evaluate a classical approach (dubbed as classic), which is a random combination of rotation, colour-jitter, and horizontal flipping. We organize the studied DAs into two groups. The first group consists of MixUp, Man, CutMix, CutOut, and Gen, while the second group includes classic, and combinations of the previous DAs with classic (dubbed as for example cls+MixUp). We also vary the probability of augmentation $p_{aug} \in \{0.1, 0.2, \ldots, 1\}$, which determines how many samples are augmented. For each DA method and augmentation probability $p_{aug}$ (which was kept fixed throughout training), we train Resnet18 (Han et al., 2019) classifiers using SGD. We use three random seeds for initialization and average the performances, as the variance is negligible. More details about our experimental setup are provided in Appendix A.

**Roughness Setup:** All used augmentations, parameters and networks are analogous to the previous setup.

**Spuriousness Setup:** In this experiment, we measure spuriousness for a data-augmentation method, via the distance of augmented data to the sets of robust and non-robust features. Following Rebuffi et al. (2021), we sample 10K images from the robust feature dataset (uniformly across classes) and 10k samples from the non-robust features dataset based on CIFAR-10.[4] These 20K images are then passed through the pretrained VGG network which measures a Perceptual Image Patch Similarity, also known as LPIPS (Zhang et al., 2018). The resulting concatenated activations are used to compute the top-100 PCA components, allowing to compare samples in a much lower dimensional space (i.e., 100 instead of $124,928$). Finally, for each augmentation method, 10K augmented images are sampled from the training set of CIFAR-10, and are passed through the pipeline composed of the LPIPS VGG network and the PCA projection computed on the original data. For each sample, we find he closest neighbor in the PCA-reduced feature space and we determine whether it belongs to the robust or non-robust sets, or to the set of augmented images (self). We then calculate the percentage of such neighbors for augmented, robust, and non-robust features. A higher percentage for a set (robust, non-robust) refers to a smaller two-sample distance to this set, and thus, to a higher similarity to this set.

## 4.2 Results

We present our results in this section, addressing the two hypotheses with results on performance-vs-robustness; and then present results on decision-function roughness and finally, data-augmentation spuriousness.

### 4.2.1 performance-vs-robustness Results

**Hypothesis 1:** We first investigate whether DAs increase adversarial robustness. In Figure 1, we plot robustness vs. performance. Compared to the no-augmentation ('noaug') baseline (red cross), in the first group (round markers) the robustness significantly degrades in MixUp (blue), Man (yellow), and CutMix (green). Robustness further slightly decreases in CutOut (purple) as the augmentation probability increases (more intense color). Gen (grey) is the only method from the first group that slightly improves robustness. From the second group (square markers), classic improves robustness by a large margin, in particular when compared to the other DAs. Concerning classification performance, the first group's performance increases for Man (yellow) and CutMix (green). For CutOut (purple), the performance increase is only small; while performance degrades strongly in Gen (gray) and slightly in MixUp (blue). In the second group, all methods significantly improve classification performance compared to noaug, with the exception of cls+Gen (grey) which, using the highest augmentation probability, performs on par with noaug. Finally, all methods show higher performance and robustness when combined with classic, compared to when applied alone. We further summarize our results in Table 2, where we summarize increases and decreases from Figure 1. The table shows how the robustness consistently decreases for group 1 (the DAs alone), but increases for group 2 (classic and combinations therewith). We thus reject hypothesis 1, that other augmentations than classic increase robustness.

---

[4] Available at `https://github.com/MadryLab/constructed-datasets`.

> **Results 1.** *We reject Hypothesis 1 by providing examples on two datasets, and using the discussed data augmentation methods, where an increased robustness is **only** achieved when the tested method is combined with classic DA. In other words, when the proposed DA is applied alone, it results in significant **increased** adversarial vulnerability.*

Table 2: Results for hypothesis 1. We provide the overall trend compared to the baseline (no augmentation) for each DA, where DAs are divided into two groups, depending on whether they are combined with a classical approach (w/Cls). We further denote whether robustness (Rob.) or accuracy (Acc.) increase slightly (↑) or significantly (↑↑), stay the same (−), or decrease slightly (↓) or significantly (↓↓). Finally, we summarize whether the effect of the augmentation probability (aug.P.) is negligible (−), small (+), or strong (++).

|  |  | Rob. | Acc. | aug.P. | w/ Cls |
|---|---|---|---|---|---|
| Group 1 | **MixUp** | ↓↓ | ↓ | ++ | No |
|  | **Man** | ↓↓ | ↑↑ | + | No |
|  | **CutMix** | ↓↓ | ↑↑ | ++ | No |
|  | **CutOut** | ↓ | ↑ | − | No |
|  | **Gen** | − | ↓↓ | ++ | No |
| Group 2 | **classic** | ↑↑ | ↑↑ | + | No |
|  | **cls+MixUp** | ↑ | ↑↑ | + | Yes |
|  | **cls+Man** | ↑ | ↑↑ | + | Yes |
|  | **cls+CutMix** | ↑ | ↑↑ | + | Yes |
|  | **cls+CutOut** | ↑↑ | ↑↑ | − | Yes |
|  | **cls+Gen** | ↑ | ↑ | ++ | Yes |

**Hypothesis 2:** We now investigate the hypothesis that the augmentation percentage does not influence generalization and robustness. We again turn to Figure 1. In the plots, the colored lines (from opaque to strong) denote the increase of the augmentation probability $p_{aug}$. The plots thus support that $p_{aug}$ has a significant effect on robustness and performance, which is more pronounced when non-classical approaches are used in isolation. In terms of robustness, compared to a baseline without DA (noaug), increasing the amount of augmentation $p_{aug}$ in the first group (round markers) significantly reduces robustness in MixUp (blue), Man (yellow), and CutMix (green), and slightly reduces in CutOut (purple). In the second group (square markers), the augmentation probability has some effect on cls+MixUp(blue) and cls+CutMix(green). This effect is however weaker than in the first group. In terms of classification performance, compared to noaug (red cross), increasing the amount of augmentation $p_{aug}$ slightly decreases the classification performance in MixUp, while in CutMix and Man the performance is improved. CutOut slightly improves performance in comparison to noaug, and Gen significantly increases performance with higher amounts. In the second group, the augmentation probability has some effect on the combinations of cls+CutMix, cls+Man, and cls+MixUp. This effect is however weaker than in the first group. The only technique that is overall, or in both groups, largely unaffected in robustness and performance from the augmentation probability is CutOut. We summarize these results again in Table 2. In short, the augmentation percentage $p_{aug}$ does have a significant influence on almost all augmentations, but in particular for DAs other than classic and when applied in isolation.

> **Results 2.** *We reject Hypothesis 2 by providing extensive results on several datasets and DAs, demonstrating the significant effect of the percentage of augmented samples on both generalisation and adversarial robustness of models, despite the use of fixed augmentation percentage choices in the literature.*

### 4.2.2 Decision-function roughness Results

To investigate the effect of the previously studied DAs on the shape of the decision surface of models, we compare our decision-function roughness measure with the risk under attack in Figure 2. As before, the two groups of augmentations exhibit different behavior. For all setups, decision-function roughness has a high correlation to vulnerability for the methods of the first group in Table 2. This suggests that methods with

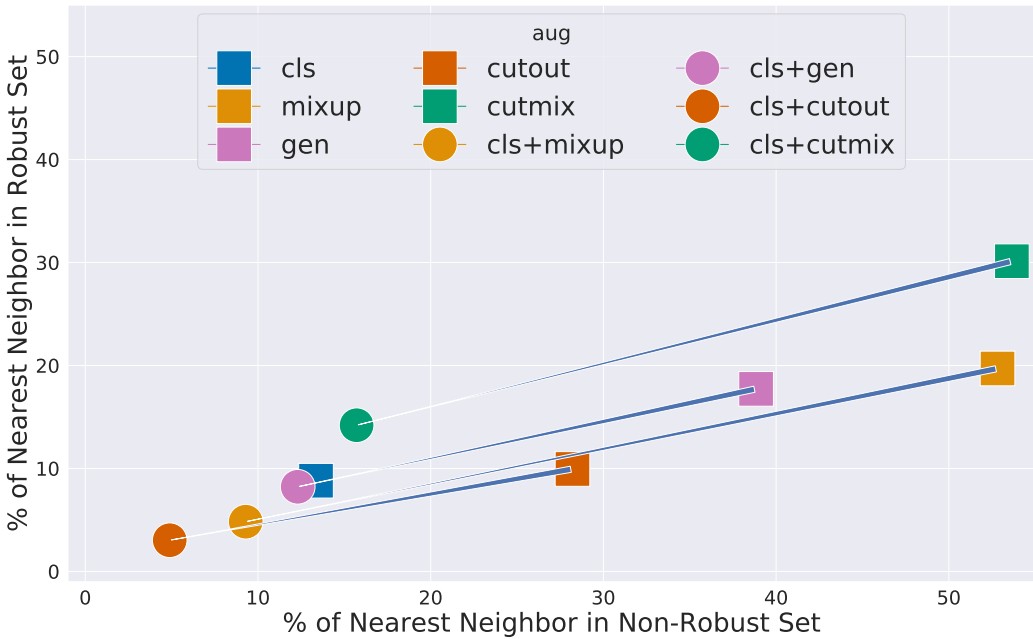

Figure 3: Results for data-augmentation spuriousness as a % of nearest neighbor of augmented data to robust and non-robust features on CIFAR10.

high decision-function roughness are vulnerable to adversarial attacks. Our result is consistent with related works (Lyu et al., 2015; Cisse et al., 2017; Sokolic et al., 2017; Cohen et al., 2019) on the relation between the shape of decision surface and adversarial vulnerability, suggesting that "rough" decision boundaries are key factors of adversarial vulnerability. In other words, we find that DAs that cause vulnerability, also induce rough decision function surfaces. Furthermore, DAs such as classic which have shown increased adversarial robustness, the percentage of augmentation does not significantly affect decision-function roughness. In contrast, DAs such as MixUp which resulted in reduced robustness are significantly affected by the augmentation percentage. Hence, such augmentations that cause rough decision functions, are more strongly affected by changes in augmentation probability.

### 4.2.3 Data-Augmentation Spuriousness Results

We plot these percentages in Figure 3. The most adversarially vulnerable DAs (MixUp (yellow), CutMix (green)) are the closest to non-robust features. Furthermore, Gen (violet) and CutOut (orange), which were in relation to the previous methods less vulnerable to adversarial attacks, are relatively further away from non-robust features. Finally, the most robust DA, classic (blue), is furthest away from the non-robust features. Additionally, once combined with classic, the augmented data increases its distance to non-robust features, which has also been reflected in their robust performance in our previous results. Using the proposed DA spuriousness, we find that DAs resulting in robust models create samples that are distant to spurious features. Comparing the distance of augmented data to robust features, which we use to put the distance to spurious features in perspective, we observe that most augmentations are relatively distant to robust features, in particular in comparison to their distance with non-robust feature. This result was to be expected, as the studied DAs do not incorporate adversarial directions in creating augmented samples, which in fact was leveraged in the generation process of 'robust features'.

## 5    Conclusion and Future Work

Recently-proposed heuristic and data-driven DA methods including MixUp (Zhang et al., 2017), CutMix (Yun et al., 2019), ManifoldMixUp (Verma et al., 2019), CutOut (DeVries & Taylor, 2017), and Diffusion Models (Ho et al., 2020) have been claimed not only to improve generalization, but also adversarial robustness. However, they have been tested only in combination with classical augmentations (like image flipping and rotation), and using a fixed fraction of augmented samples. This questions whether robustness is really induced by the newly-proposed DA strategies themselves, or it is instead induced mostly by classical augmentations and specific choices of the augmentation probability. In this work, we shed light on this issue by proposing an evaluation framework that helps decouple the impact of such factors on both accuracy and robustness, through the definition of different metrics that characterize the augmented manifold. We re-evaluate recently-proposed heuristic and data-driven DAs using our framework, and find contradictory evidence when compared to prior work. In particular, our extensive empirical analysis on the aforementioned DA methods has shown that: (i) such recently-proposed DA methods only improve adversarial robustness when combined with classical augmentations; (ii) they worsen adversarial robustness if not combined with classical augmentations; and (iii) adversarial robustness is significantly affected by augmentation probability. This demands for future work aimed to rethink not only the evaluation but also the development of novel DA methods for adversarial robustness, and we firmly believe that our work provides a significant first step in this direction.

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

## A  Experimental Setup

### A.1  Training Setup

Image classification experiments are carried out using a ResNet18 He et al. (2016). The ResNet model was trained using SGD with a momentum of 0.9, and weight-decay with penalty coefficient of $5e - 4$, with batch size of 128. Each classifier was trained for 200 epochs and learning rate schedule was used with initial value of 0.1, which was reduced twice by a factor of 10 every 80 epochs.

#### A.1.1  Data Augmentations

**MixUp**: the mixing parameter $\lambda$ was drawn from $\mathcal{B}(1,1)$. **Manifold-MixUp**: the mixing parameter $\lambda$ was drawn from $\mathcal{B}(2,2)$. The eligible layers on CIFAR10 was set to $\mathcal{S} = \{0,1,2\}$, while on CIFAR100 $\mathcal{S} = \{0,1,2,3\}$. **CutMix**: the mixing parameter $\lambda$ was drawn from $\mathcal{B}(1,1)$. Bounding boxes have been randomly chosen for the cutout operation, with cut ratio of $\sqrt{1-\lambda}$. **CutOut**: with 1 hole and the length of 16 pixel has been used. **Classic**: is a random combination of Random Cropping, Horizontal Flipping, Colour jittering, and Random Rotation. Random Cropping is done with the padding of 4 and size of 32. For Colour jittering, brightness, contrast, and saturation factors have been changed by a random amount chosen uniformly from $[0.75, 1.25]$. **Gen.**: was utilized using samples generated by a Denoising Diffusion Probabilistic model (Ho et al., 2020) trained on CIFAR10. These samples have been released as CIFAR5M dataset Nakkiran et al. (2020)[5]. Due to the computational complexity of DDPM, and lack of availability of such generated dataset on CIFAR100, we opted to only use Gen. on CIFAR10. When a sample is chosen to be augmented by Gen., we replace the original sample by a randomly chosen example of the same class from CIFAR5M. **Combinations with Classic**: In all combination experiments, in addition to the target augmentation (e.g, MixUp), samples have been additionally augmented with Classic, with augmentation probability of 0.5.

#### A.1.2  Adversarial Attack Setup

We carry out 4 different untargeted PGD attacks with $L_2$ norm and 4 different untargeted PGD attacks with $L_\infty$ norm. For PGD attacks with $L_2$ norm, we use perturbation sizes of $\mathrm{PGD}_\epsilon \in \{0.01, 0.1, 0.5, 1\}$, and for PGD attacks with $L_\infty$ norm perturbation sizes of $\mathrm{PGD}_\epsilon \in \{\frac{1}{255}, \frac{2}{255}, \frac{4}{255}, \frac{8}{255}\}$ have been utilized. All attacks have been conducted with the step size that is $\frac{1}{5}$ of the perturbation size ($\mathrm{PGD}_\alpha = \frac{\mathrm{PGD}_\epsilon}{5}$), and with the number of iterations of 100.

### A.2  Implementation

All experiments have been implemented in python using `pytorch lib` Paszke et al. (2019). The adversarial attacks are done using the `robustness lib` Engstrom et al. (2019).

## B  Extended Results

### B.1  Augmentation Probability

We provide a summary for the influence of augmentation probability on robustness and performance for different augmentations in Table 3. As can be seen, while classic (Cls) and CutOut are less sensitive to the augmentation probability, MixUp, CutMix demonstrate high sensitivity in both. Also Gen. is more sensitive in performance and not robustness, while Manifold-MixUp is more sensitive in terms of robustness, and less sensitive in terms of performance. This trend is similar in cases with, and without combination with classic.

---

[5]This dataset is publicly available here: `https://github.com/preetum/cifar5m`

Table 3: Extended summary of the influence of augmentation probability on robustness and performance. Comb. w/ Cls: combined with classic. Man.: Manifold-MixUp. Cls.: Classic. Gen.: generative model.

|  | Performance | Robustness | |
|---|---|---|---|
| **MixUp** | high | high | |
| **Man.** | low | high | |
| **CutMix** | high | high | Single |
| **CutOut** | low | low | |
| **Gen.** | high | low | |
| **Cls** | low | low | |
| **MixUp+Cls** | low | high | |
| **Man.+Cls** | low | high | |
| **CutMix+Cls** | low | high | Comb. w/ Cls |
| **CutOut+Cls** | low | low | |
| **Gen.+Cls** | high | low | |

Table 4: Extended summary of the results. Stress: Prediction-change stress (high indicates adversarial vulnerability). RUA: Risk under attack (high indicates adversarial vulnerability). Dist. to NRS: Distance to non.robust set (low indicates adversarial vulnerability). Imp. (Pr): Impact of augmentation probability on robustness. Perf.: Performance, inverse of misclassification risk. low:↓↓. high:↑↑. medium:↑↓. Man.: Manifold-MixUp. Gen.: Generative model. Cls: Classic.

|  | Roughness | RUA | Dist. to NRS | Imp. (Pr) | Perf. |
|---|---|---|---|---|---|
| **Cls** | ↓↓ | ↓↓ | high | low | ↑↑ |
| **MixUp** | ↑↑ | ↑↑ | low | high | ↓↓ |
| **CutMix** | ↑↑ | ↑↑ | low | high | ↑↓ |
| **CutOut** | ↑↓ | ↑↑ | med | low | ↑↓ |
| **Gen.** | ↑↓ | ↓↓ | med | low | ↓↓ |
| **Man.** | ↑↓ | ↑↓ | N/A | high | ↑↓ |

## C Tables

### C.1 Distance to Robust and Non-robust Features

Table 5: The percentage of nearest neighbors of the augmented data on CIFAR10, in each set.

|  | Self | Robust Features | Non-rob. Features |
|---|---|---|---|
| **Cls** | 77.83 | 8.81 | 13.36 |
| **MixUp** | 27.47 | 19.71 | 52.82 |
| **Gen** | 43.43 | 17.73 | 38.84 |
| **CutOut** | 61.84 | 9.95 | 28.21 |
| **CutMix** | 16.22 | 30.13 | 53.65 |
| **Cls+MixUp** | 85.88 | 4.83 | 9.29 |
| **Cls+Gen** | 79.48 | 8.22 | 12.30 |
| **Cls+CutOut** | 92.08 | 3.03 | 4.89 |
| **Cls+CutMix** | 70.09 | 14.20 | 15.71 |

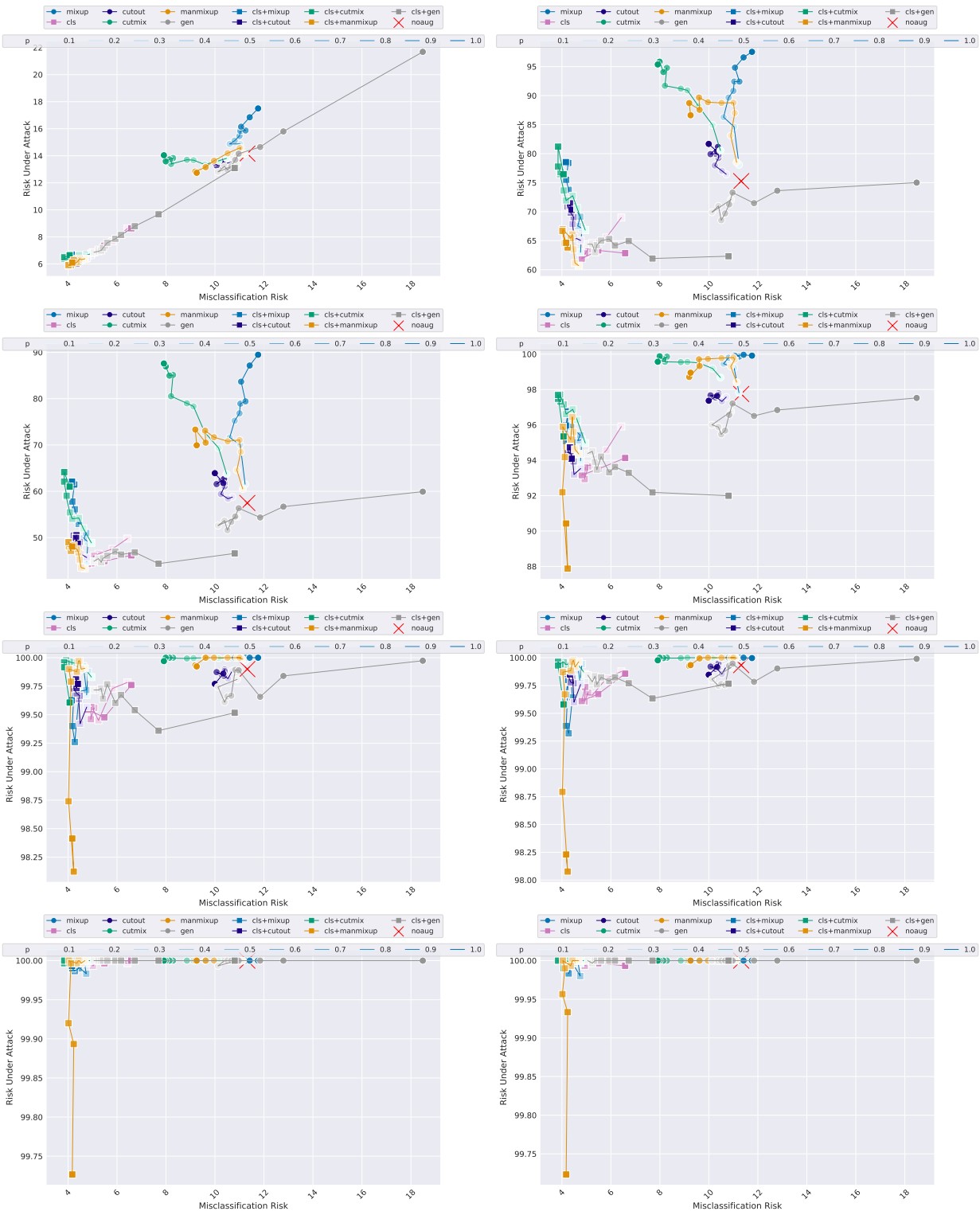

Figure 4: Robustness vs. Performance on CIFAR10. first column: risk under attack ($L_2$) VS. misclassification risk second column: risk under attack ($L_\infty$) VS. misclassification risk Perturbation sizes for $L_2$ PGD attacks are shown in each row: $\{0.01, 0.1, 0.5, 1\}$. Perturbation sizes for $L_\infty$ PGD attacks are shown in each row: $\{\frac{1}{255}, \frac{2}{255}, \frac{4}{255}, \frac{8}{255}\}$.

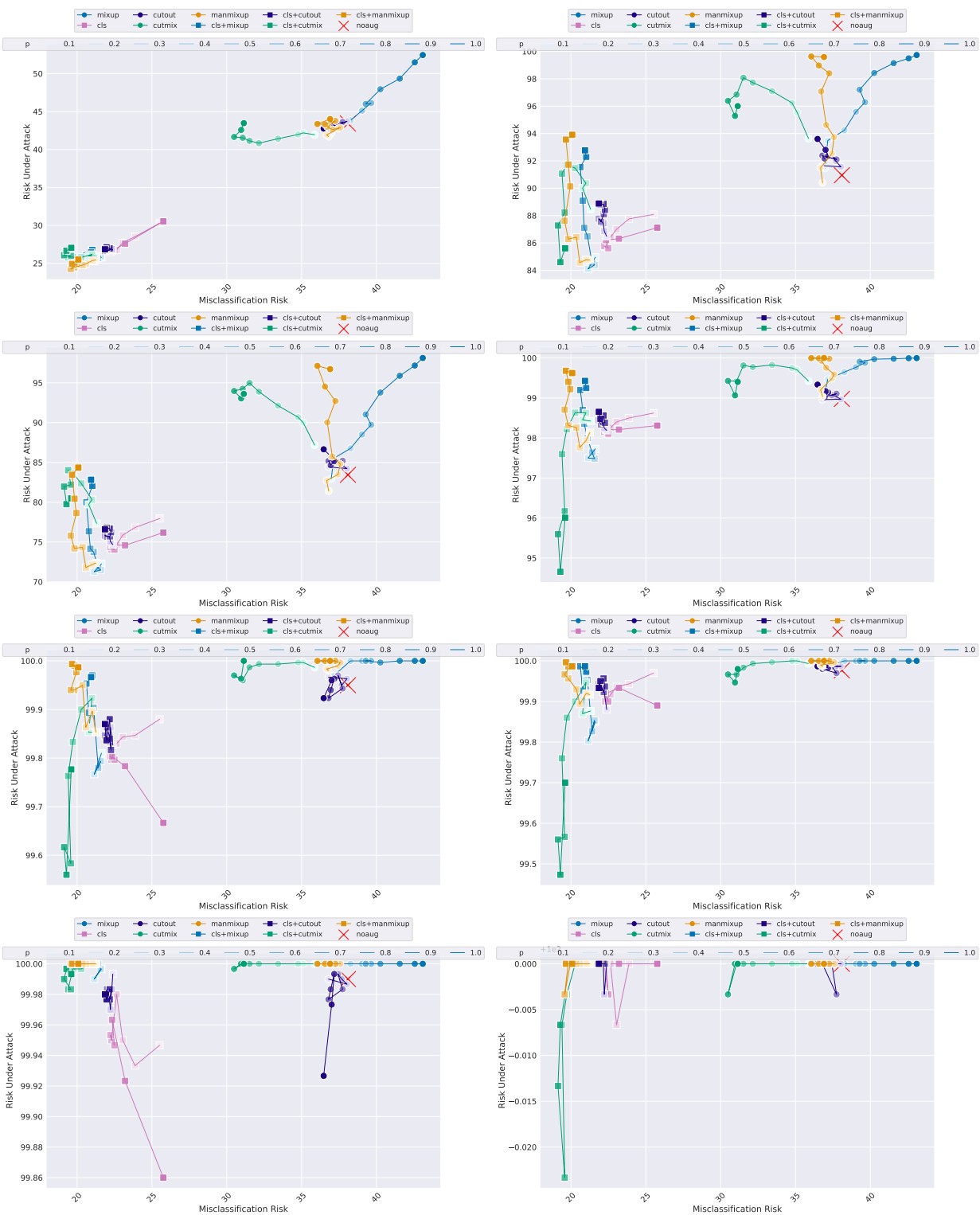

Figure 5: Robustness vs. Performance on CIFAR100. first column: risk under attack ($L_2$) VS. misclassification risk second column: risk under attack ($L_\infty$) VS. misclassification risk Perturbation sizes for $L_2$ PGD attacks are shown in each row: $\{0.01, 0.1, 0.5, 1\}$. Perturbation sizes for $L_\infty$ PGD attacks are shown in each row: $\{\frac{1}{255}, \frac{2}{255}, \frac{4}{255}, \frac{8}{255}\}$.

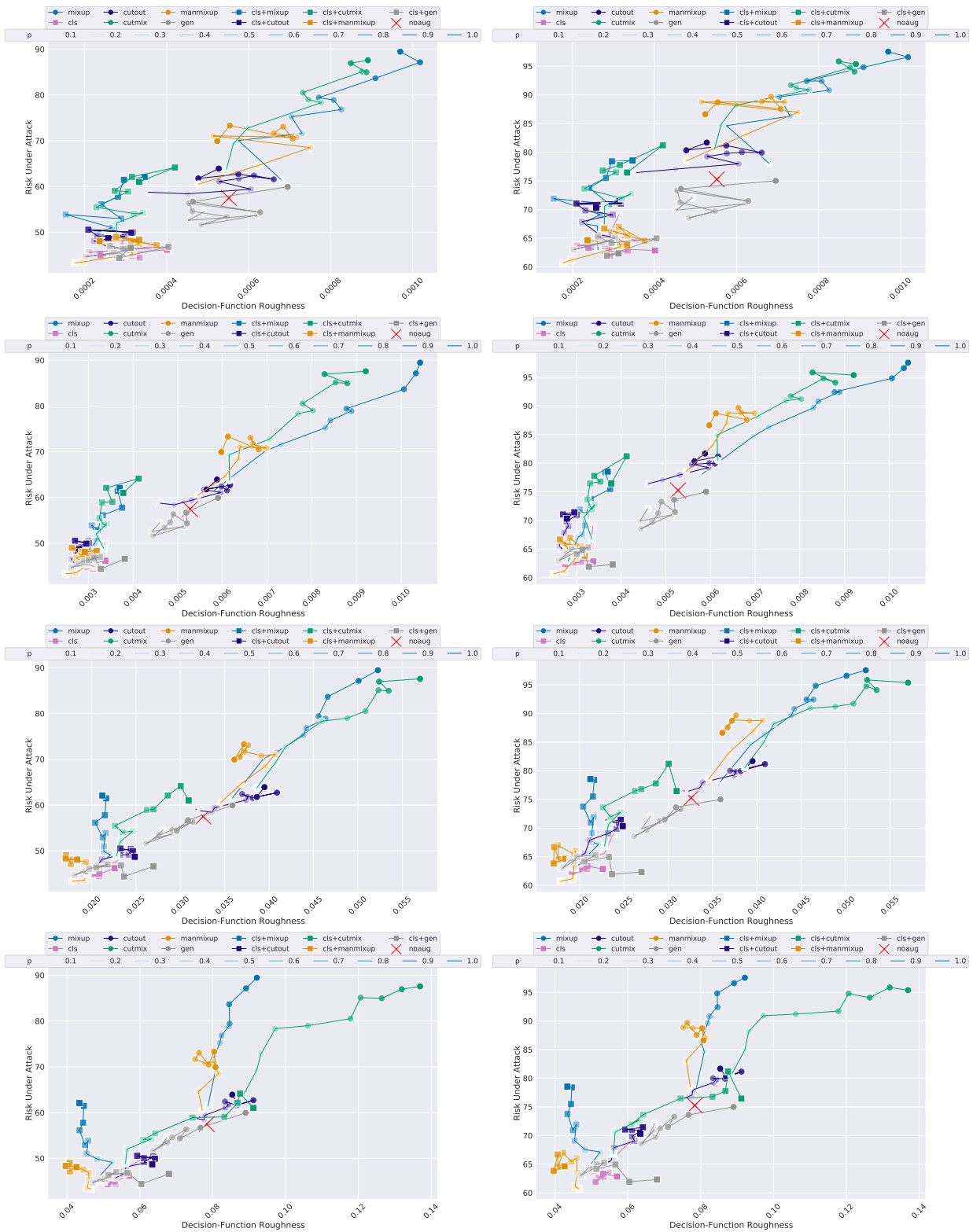

Figure 6: Decision-function roughness vs. robustness on CIFAR10. first column: roughness vs. risk under attack ($L_2$) second column: roughness vs. risk under attack ($L_\infty$) rows: $\epsilon_{stress} = \{0.01, 0.1, 0.5, 1, 2\}$. PGD attacks with $L_2$ and $L_\infty$ used perturbation size of 0.1, and $\frac{1}{255}$, respectively.

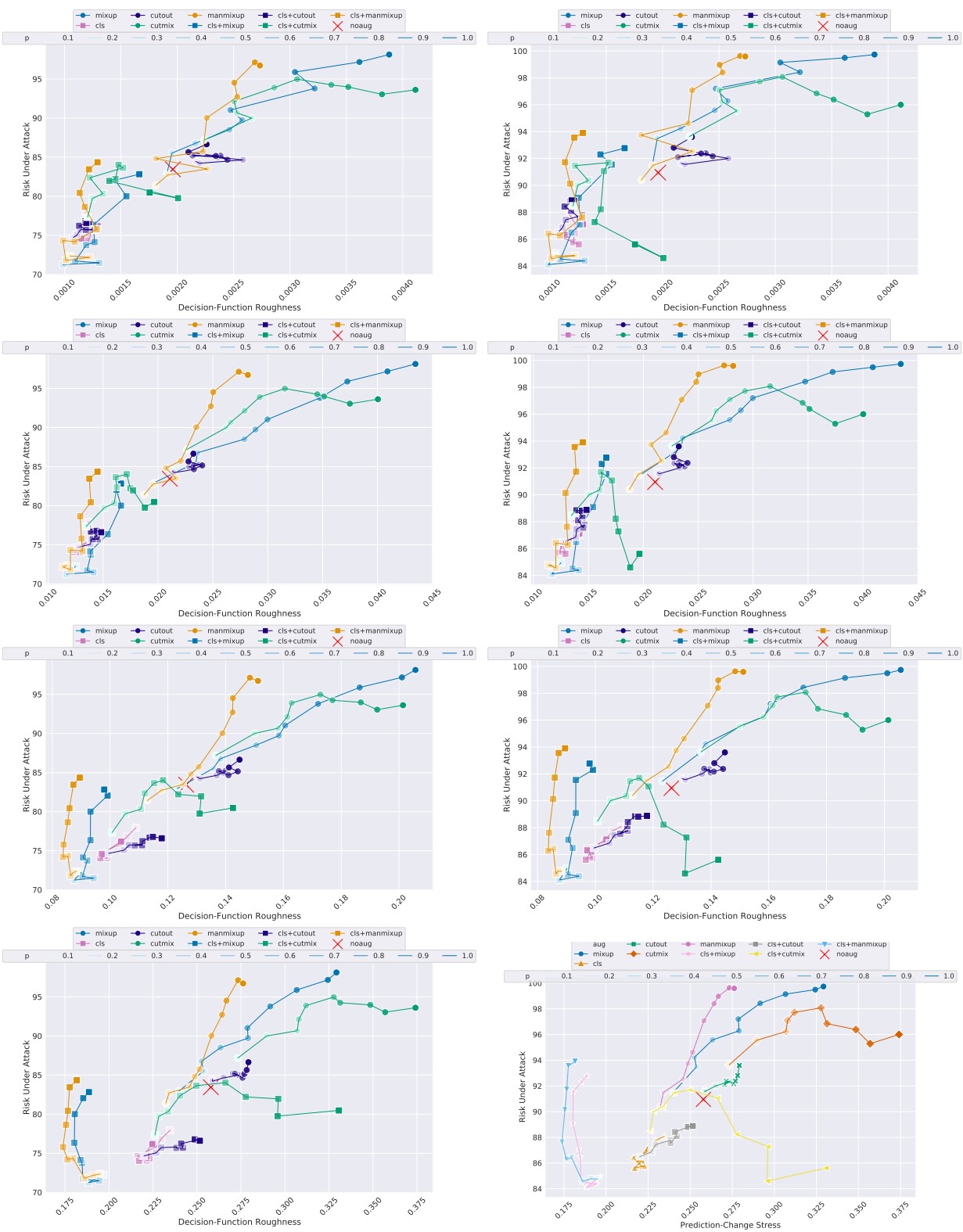

Figure 7: Decision-function roughness vs. robustness on CIFAR100. first column: roughness vs. risk under attack ($L_2$) second column: roughness vs. risk under attack ($L_\infty$) rows: $\epsilon_{stress} = \{0.01, 0.1, 0.5, 1, 2\}$. PGD attacks with $L_2$ and $L_\infty$ used perturbation size of 0.1, and $\frac{1}{255}$, respectively.

