# OpenReview forum: "Evaluation Pitfalls in Data Augmentation for Adversarial Robustness"
_TMLR — Rejected by TMLR_

### Review · Reviewer_yxZr · 2022-06-11

**Summary Of Contributions:**

The authors observe that, while existing work has claimed that a set of "heuristic" and "data-driven" data augmentation methods have improved adversarial robustness, the studies have i) coupled said methods with "classical" data augmentation methods, and ii) fixed the choice for the # of samples that are augmented (augmentation probability choice).

In experiments, they find the choices of (i) and (ii) are consequential in affecting the results, and find interesting interactions elucidating why these consequences were not observed in previous work, i.e. coupling with "classical" data augmentation methods leads to decreased sensitivity to augmentation probability choice. Overall, according to the authors' assessment of existing work, this work contributes results which provide a more nuanced view on the relationship between data augmentations and adversarial robustness.

**Broader Impact Concerns:**

No concerns regarding this work in particular.

**Requested Changes:**

None to mention.

**Strengths And Weaknesses:**

Strengths:
* Well-written and solid presentation.
* Thoughtful evaluation experimental setup.

Weaknesses:
* Solely considered PGD for adversarial attack (though did consider different norm constraints). To be clear, I don't consider this a significant issue, particularly given a) the evaluated models are not designed with the intention of obfuscating gradients and b) none of the evaluated models appear overfit to a particular attack approach. Still, it should be noted that recent work [1] has pointed to not evaluating on a diverse set of attacks as a common pitfall in robustness claims which did not stand the test of time. Suggested additions are listed below,
  * The authors tested 4 different epsilons per norm, but it would be interesting if for each example, the minimum-sized perturbation which is adversarial (changes output class) were found, and the median/mean +/- absolute/standard deviation were reported (see [2] for reference). While there exist attacks which jointly minimize the perturbation size while maximizing the adversarial criterion [3], note that finding the minimum-sized successful perturbation can also be done with PGD by performing a binary search over epsilon [4].
  * Regarding alternative attacks, note that in [2], the authors observed that a decision-based attack [5] outperformed PGD-L2 in attacking a network trained with PGD-Linf. While I would expect that if the same experiment were performed with the models assessed here, the aforementioned observation would not follow, it is possible that PGD may not outperform all other attacks in all scenarios. Verification that the main results were not computed using suboptimal attacks would be beneficial for the reader. A number of alternative attacks exist, many of which are implemented in open-sourced libraries; for reference, see Table 1 in [2] and sec. 4.3-4.5 in [1].
  * Given the particular nature of the models analyzed in the study, it would be interesting for the authors to investigate spatial transform attacks [6]. Here, the attacks are constrained by the class of allowable transformations, as opposed to a fixed perturbation budget quantified by Lp-norm. Notably,  the spatial transforms considered (rotation, translation) can be seen as "classical data augmentations", and thereby may provide an additional insightful angle for the study.

Nitpicks:
- "A DA technique is retained useful..." wasn't entirely clear.

[1]: Carlini, Nicholas, et al. "On evaluating adversarial robustness." arXiv preprint arXiv:1902.06705 (2019).
[2]: Schott, Lukas, et al. "Towards the first adversarially robust neural network model on MNIST." International Conference on Learning Representations. 2018.
[3]: Carlini, Nicholas, and David Wagner. "Towards evaluating the robustness of neural networks." 2017 ieee symposium on security and privacy (sp). IEEE, 2017.
[4]: https://github.com/bethgelab/foolbox/blob/v2/foolbox/v1/attacks/iterative_projected_gradient.py#L96.
[5]: Brendel, Wieland, Jonas Rauber, and Matthias Bethge. "Decision-Based Adversarial Attacks: Reliable Attacks Against Black-Box Machine Learning Models." International Conference on Learning Representations. 2018.
[6]: Engstrom, Logan, et al. "Exploring the landscape of spatial robustness." International Conference on Machine Learning. PMLR, 2019.

---

> ### Comment · Action_Editors · 2022-06-12
> **Provide more details regarding missing attack evaluation**
>
> Thank you for the review. Could you expand a bit on specific attacks which you think would be relevant here? It will help the authors understand which aspects of robustness are not well measured by their evaluation because it only uses PGD -- and how to improve their current evaluation to better characterize robustness.

---

> > ### Comment · Reviewer_yxZr · 2022-06-12
> > **Edited**
> >
> > Thanks for the feedback, provided additional details.

---

### Review · Reviewer_oLZK · 2022-06-12

**Summary Of Contributions:**

This paper studies the effect of data augmentation choices on the adversarial robustness of naturally trained (i.e., not adversarially trained) models. Specifically, the paper proposes a framework that disentangles the effect of different augmentations by ablating the augmentation pipeline and recording three different metrics, each meant to capture an aspect of adversarial vulnerability. The paper uses the framework to test two hypotheses floated by prior works: that new data augmentation methods increase adversarial robustness separately from classical methods, and the percentage of augmented samples does not significantly affect generalization and robustness.

**Requested Changes:**

I think accepting this paper would require all of the changes listed above to be made. More specifically:

- Making the hypotheses quantitative and precise
- Re-arranging sections to introduce terms before they are used
- Thorough proofreading, including removing mathiness
- Replacing the arrows in the tables with the corresponding actual values rather than hiding it behind the somewhat opaque terminology "significant"
- Running the results several times in order to get error bars, and presenting these error bars in the graphs and tables

**Strengths And Weaknesses:**

Strengths:

- The paper performs a thorough ablation study that includes a large variety of data augmentations.
- The paper makes testable hypotheses rather than just blindly exploring the landscape of data augmentations, and convincingly refutes the hypotheses. I'm aware that TMLR does not put much weight on significance or novelty but I think that the findings in the paper are both interesting and novel.
- The paper clearly states and summarizes its findings in bulleted lists
- The paper precisely defines the notation and terminology it uses, which in turn makes the claims more precise.

Weaknesses:

- The hypotheses, in my view, are too broad and vague. For example:
---- Hypothesis 1 is "Newly-proposed heuristic and generative DAs increase adversarial robustness besides classical augmentations." First of all, the hypothesis is confusingly worded (in particular, the use of "besides" here is imprecise). Second, the line between classical and heuristic DA seems vague to me: why is CutOut heuristic rather than standard DA?
---- Hypothesis 2 is even more confusing---what does it mean to "significantly influence" generalization and adversarial robustness? Also, as stated the hypothesis is clearly false since at p = 0 there is no data augmentation, and at p = 1 there is full data augmentation (and the accuracy gap between these two cases is significant). If possible, the hypothesis should be a more precise claim, for example: "When p > 0.5, the value of p will be uncorrelated [e.g., wrt Pearson r correlation]  with the accuracy of the corresponding model)

- I also think the paper should be restructured a bit: currently, for example, "data augmentation" is defined on page 4 despite being used tens of times earlier in the text.

- The paper itself could use a more thorough proofreading: there are many grammar errors that hinder reading significantly. There are also many places where math should be replaced by (or at least supplemented with) a plain-english definition of a key term (for example, Equation 2).

- The presentation of experimental results could also be significantly improved:
--- The arrows in tables should be replaced with actual numbers that include confidence intervals so that the significance can be judged directly by the reader
--- The plots are in a very confusing format: the colors are not consistent, Figure 3 looks out of place, and the axis labels are too small.

- In general, the entire paper seems to lack error bars around any of the results.

---

### Review · Reviewer_CRr7 · 2022-06-13

**Summary Of Contributions:**

This paper studies how data augmentation affects adversarial robustness in image classification models. The authors study classical data augmentation techniques (such as flipping and rotation) as well as recently proposed new data augmentation methods such as Mixup, CutOut, etc. The authors empirically show that on CIFAR-10 and CIFAR-100 models, new data augmentation methods do not necessarily improve model robustness, and in some cases robustness even descries; the hypothesis that newly proposed data augmentation improves robustness seems not hold, and the percentage of augmented samples also matters for robustness.

**Requested Changes:**

Requested changes:

1. Study the impact on robustness in different model architecutures, such as the recently popular vision transformers and MLP based architecutures. The conclusions presented in this paper may be different in these models.

2. Include experiments on ImageNet for a more convincing empirical study.

3. Add study on the impact of data augmentation in adversarially trained models, as well as the model behavior under different perturbation radii.



**Strengths And Weaknesses:**

Strengths:

1. The results presented in this paper were not thoroughly demonstrated in any existing works and are helpful for understanding how to obtain a robust model.

2. Different perspectives of robustness are considered, including decision-function roughness and Data-Augmentation Spuriousness.

3. Very detailed results with a wide range of perturbation probability p from 0.1 to 1.0 are considered, which essentially controls the percentage of augmented samples and its impact was not studied in detail in existing works.

Weaknesses:

1. In the case of obtaining adversarially robust models, I think adversarial training is essential, and it is more important to investigate the impact of data augmentation with adversarial training. It seems most results in this paper are produced without adversarial training, and I feel it is important to repeat all experiments under adversarial training settings as well.

2. Since this paper focuses on empirical study of data augmentation, I think it is necessary to include more model architectures and datasets into the experiments to observe a consistent conclusion. Especially, ImageNet models are essential for this type of empirical study because CIFAR datasets are too small and not representative.

---

### Decision · Action_Editors · 2022-07-18

**Recommendation:** Reject

**Comment:**

I would like to thank the authors for their submission to TMLR. I hope the following feedback, together with the reviews and discussion, will provide some insights in the outcome and a way forward for the paper.

## Criteria

I assessed the submission and reviews based on the two criteria listed by TMLR.

> Are the claims made in the submission supported by accurate, convincing and clear evidence?

I did not find any issues with the submission's claims in the discussions with Reviewer CRr7 and Reviewer yxZr. However, the discussion with Reviewer oLZK led to unresolved disagreements on:
* the hypotheses that underly the proposed work. Specifically, the reviewer pointed out that there is no clear distinction “between newly proposed" and "classical" data augmentation techniques.
* the need for error bars to accompany all numerical results

Both of these are limitations of the work. I am also concerned that both issues would require additional experimentation to obtain the data needed - and would thus not be a good fit for a "minor revision" outcome.


> Would at least some individuals in TMLR's audience be interested in the findings of this paper?

All 3 reviewers agreed that the paper reports findings that would be interesting to the TMLR audience:
* Reviewer CRr7 describes the findings as helpful in understanding how to obtain a robust model.
* Reviewer oLZK describes the findings as significant.
* Reviewer yxZr write "this work contributes results which provide a more nuanced view on the relationship between data augmentations and adversarial robustness"

## Overall decision

I would like to thank the authors for engaging with reviewers in a discussion following the reviews being posted. While I found that the issues raised by Reviewer oLZK are not sufficiently clearly addressed in the author response for the paper to pass the first criterion, the second criterion is clearly satisfied. Thus, I encourage the authors to further refine the hypotheses that underly their work (perhaps through a continued discussion with Reviewer oLZK on OpenReview) and to add the experimental data missing (error bars and any new results called for by refined hypotheses). I believe the resulting paper should then be reviewed again and considered closely for publication at TMLR.

## Other comments

I did not consider the following in my decision but thought they are still important points brought up by the reviewers.

* Reviewer CRr7 highlights the lack of consideration of adversarial training as being an important area for improvement. In particular, they cite recent work on adversarial training which incorporates data augmentations as a motivation for studying the interplay between the two. Also note that Reviewer oLZK has provided some feedback on how to run experiments with adversarial training at a moderate expense in terms of hyper parameter tuning.
* Finally, if the authors are to revise their hypotheses, they may find it useful to consider attacks beyond PGD - as discussed with Reviewer yxZr.